# Forest and Landscape Restoration: A Review Emphasizing Principles, Concepts, and Practices

**Ricardo Gomes César** [1,*], **Loren Belei** [2], **Carolina Giudice Badari** [1], **Ricardo A. G. Viani** [3], **Victoria Gutierrez** [4], **Robin L. Chazdon** [5], **Pedro H. S. Brancalion** [1] and **Carla Morsello** [6]

1   Department of Forestry Sciences, 'Luiz de Queiroz' School of Agriculture (ESALQ), University of São Paulo, Av. Pádua Dias, 11, Piracicaba 13418-900, Brazil; carolgbadari@alumni.usp.br (C.G.B.); pedrob@usp.br (P.H.S.B.)
2   Graduate Program in Environmental Science, Institute of Energy and Environment, University of São Paulo, Av. Prof. Luciano Gualberto, 1289, São Paulo 05508-900, Brazil; loren.belei@usp.br
3   Department of Biotechnology and Plant and Animal Production, Federal University of São Carlos, Agraria Science Centre (CCA), Rodovia Anhanguera, Km 174, Araras 13604-900, Brazil; viani@ufscar.br
4   Commonland, Kraanspoor 26, 1033 SE Amsterdam, The Netherlands; victoria.gutierrez@commonland.com
5   Tropical Forests and People Research Centre, University of the Sunshine Coast, Maroochydore DC, Sunshine Coast, QLD 4558, Australia; rchazdon@usc.edu.au
6   School of Arts, Sciences and Humanities, University of São Paulo, Arlindo Bettio Street, 1000, São Paulo 03828-000, Brazil; morsello@usp.br
*   Correspondence: ricardogoce@yahoo.com.br

**Abstract:** Forest and Landscape Restoration (FLR) is considered worldwide as a powerful approach to recover ecological functionality and to improve human well-being in degraded and deforested landscapes. The literature produced by FLR programs could be a valuable tool to understand how they align with the existing principles of FLR. We conducted a systematic qualitative review to identify the main FLR concepts and definitions adopted in the literature from 1980 to 2017 and the underlying actions commonly suggested to enable FLR implementation. We identified three domains and 12 main associated principles—(i) Project management and governance domain contains five principles: (a) Landscape scale, (b) Prioritization, (c) Legal and normative compliance, (d) Participation, (e) Adaptive management; (ii) Human aspect domain with four principles: (a) Enhance livelihoods, (b) Inclusiveness and equity, (c) Economic diversification, (d) Capacity building; (iii) Ecological Aspects domain with three principles: (a) Biodiversity conservation, (b) Landscape heterogeneity and connectivity, (c) Provision of ecosystem goods and services. Our results showcase variations in FLR principles and how they are linked with practice, especially regarding the lack of social aspects in FLR projects. Finally, we provide a starting point for future tools aiming to improve guidance frameworks for FLR.

**Keywords:** literature review; forest restoration; human dimension of restoration; ecosystem services; landscape ecology; project management

## 1. Introduction

Forest and Landscape Restoration (FLR) emerged in 2000 as a novel approach to regain ecological functionality and strengthen human well-being in deforested and degraded areas [1,2]. The FLR approach expanded from ecological restoration and from reflection upon failures in conservation and forest management approaches, and addresses interventions to recover or conserve native ecosystems. These interventions include farming and other initiatives to improve outcomes for local livelihoods, ecosystem services (ES), and biodiversity conservation at the landscape scale [3]. More recently, FLR has been included within the umbrella of "Nature-based Solutions", and is aligned with other approaches to solve complex socio-environmental problems [4].

Forest and Landscape Restoration aims to better address the often-neglected human dimensions of restoration [5–7]. Although the human spectrum of restoration is important

for stakeholder engagement, and thus for long-term restoration success [8], a systematic review of restoration monitoring found that 94% of the articles addressed the ecological aspects of restoration, while only 3.5% considered socio-economic ones [7]. The relatively few studies worldwide on the socio-economic aspects of restoration—when compared to those based on ecological aspects—focused on specific issues such as local community engagement, resource investments, job and income generation [5,7,9,10], or psychological outcomes (e.g., life satisfaction or the psychological benefits of restoration activities [11,12]).

Ideally, a broad set of human dimensions and socio-economic outcomes should be evaluated and integrated into restoration projects to ensure and assess achievements [8,13]. Such holistic overview is especially necessary for FLR because this approach recognizes the need to address the drivers of deforestation and land degradation. Moreover, FLR often depends on improving the long-term sustainability of production systems that may have negative short-term impacts on local livelihoods [14], especially where land tenure is insecure [15]. Without the active involvement of local people and other stakeholders, restoration may fail to fulfill the expected goals or lead to unintended negative consequences [8,16–18].

Evaluation of restoration initiatives focuses primarily on the ecological and biophysical outcomes of restoration [7]. More recently, a growing body of literature indicates the importance of human dimensions, such as socio-economic aspects and stakeholder engagement aspects for long-term restoration success [8,13,19]. The lack of appropriate consideration of key factors underlying restoration success may result, among other things, from the absence of a shared set of guiding principles and lack of interdisciplinary approaches. Despite several documents conceptualizing FLR and its principles [2,20], few systematic efforts have identified evidence-based principles of FLR activities that have been implemented on the ground [21–23]. A review of both the scientific and the practitioners' literature ("grey literature", such as case studies, reports and policy briefs) could assist the identification of existing concepts and practices associated with the ecological and human aspects of FLR, and ultimately offer critical guidance to the implementation of the >200 Mha of restoration commitments made to the Bonn Challenge and of the upcoming United Nations Decade on Ecosystem Restoration (2021–2030).

Since FLR is a relatively recent restoration approach encompassing multiple human, ecological and economic dimensions, its principles and strategies are being constantly reviewed and refined [8,24–26], similarly to key underlying attributes of FLR such as such gender equality [27], land tenure [18], funding [28], and definitions [1]. There is a vast and still-growing literature of case studies of FLR projects that assess the implementation of principles, identify common challenges and make recommendations [13,18,21,26,29–33]. However, the holistic and complex principles of FLR defined in the literature are challenging to implement in practice [33]. Here, instead of evaluating FLR projects on the ground, we assess the FLR principles and criteria in the literature published by practitioners and researchers. We conducted a systematic qualitative review to identify the main FLR concepts and definitions adopted in the academic and "practitioners" literature, and the underlying strategies commonly suggested to enable FLR implementation in different socio-ecological contexts. More specifically, we identified the main FLR principles in the literature, identified gaps, and provided recommendations based on existing established principles. Our analysis uncovered some main FLR principles in the literature that we organized across three domains and 12 main associated principles—(i) Project management and governance domain contains five principles: (a) Landscape scale, (b) Prioritization, (c) Legal and normative compliance, (d) Participation, (e) Adaptive management; (ii) Human aspect domain has four principles: (a) Enhance livelihoods, (b) Inclusiveness and equity, (c) Economic diversification, (d) Capacity building; (iii) Ecological Aspects domain has three principles: (a) Biodiversity conservation, (b) Landscape heterogeneity and connectivity, (c) Provision of ecosystem goods and services. The main gaps in the literature include a lack of reference to the socio-economic and monitoring aspects of FLR, although recently these subjects have been increasingly addressed.

## 2. Materials and Methods

We conducted a systematic qualitative review of the literature, i.e., an explicit, repeatable and standardized procedure to identify, select and code textual data [34]. We chose a qualitative review since we aimed to understand the conceptual basis of FLR among researchers and practitioners amidst the proliferation of terms used in this practice. To do so, we adopted Thematic Analysis, a method appropriate to summarize features of large qualitative data sets [35], through the identification, organization, analysis, and report of patterns and themes [36]. The methodology was organized into four steps.

### 2.1. Collecting FLR Documents

We systematically collected FLR documents to increase transparency, guarantee a diversity of perspectives, and reduce sampling bias [37,38]. Because we were interested in conceptual aspects (e.g., how people and organizations define FLR) and qualitative evidence on FLR characteristics/implementation, we did not consider FLR effectiveness or hypotheses regarding its effectiveness, as in conventional quantitative/"rationalist" systematic reviews or counterfactual analyses [39]. Instead, we systematically identified and coded concepts and/or arguments in the literature and, when needed (e.g., different words were used), harmonized the terminology [40]. We followed the overall guidelines for data validity and reliability, such as code generation, revision, codebook recording, researcher triangulation, peer debriefing, description of terms, and prolonged engagement with data [41].

We reviewed the scientific literature (journal articles, books, and book chapters), based on bibliographic searches from 1980 to 2017 in Scopus (www.scopus.com) and in Web of Science (WoS, apps.webofknowledge.com). We sampled the literature until 2017 because this review was carried out in early 2018. In Scopus, we selected documents of all types and searched for terms in the title, abstract or keywords. In WoS, we selected "Full Collection" and searched in the keywords. In both databases, we searched for terms: "forest" (OR woodland OR trees OR plantation OR rainforest), "restoration" (OR revegetation, forestation, afforestation, rehabilitation) AND "landscape", and then the Boolean combinations of three words: (i) "landscape restor*" AND forest (OR woodland OR trees OR plantation OR rainforest); (ii) "forest restor*" AND landscape, (iii) "forest landscape" AND restoration (OR revegetation OR forestation OR afforestation OR rehabilitation). Searches returned 1532 documents. We excluded duplicates using EPPI-Reviewer® v.4 (eppi.ioe.ac.uk), a web-based software for managing and analyzing review data, ending up with 843 publications. We then screened titles and abstracts of publications based on their consideration of FLR and social or ecological aspects of restoration, resulting in 118 scientific papers. We then fully read these papers and, based on the same inclusion criteria, we ended up with 94 publications (Supplementary Material 1). We recognize that this type of search may miss documents that use terms such as "community-based forest management" or "sustainable land management", which may be very similar to FLR, but we chose to look only for sources that claim to apply FLR. We also recognize that our search terms only encompass sources in English language, and that the interpretation of FLR concepts may vary among countries and groups [42,43].

We then reviewed the practitioners' "grey" literature (e.g., books, brochures, policy briefs) produced by FLR initiatives. We first identified key international organizations and initiatives working with FLR through a non-systematic web-based search, and then submitted our list to three FLR experts (two researchers and an NGO member) to check for omissions, resulting in a list of 31 initiatives (Supplementary Material 2). Because we wanted to capture organizational views instead of incorporating all available documents, while reducing selection bias, we standardized the sampling procedure in Google (www.google.com). Our aim was not to identify all initiatives and grey literature produced by practitioners; therefore, we recognize that our method may have neglected the grey literature of a few organizations (e.g., WWF), however, the work of such organizations is found in the scientific literature sampled [27,44,45]. Across all 31 organization websites,

we searched for Portable Document Format (pdf) documents available online. We began adopting the same term list used for searching the scientific literature. However, as this restricted procedure did not return useful documents, we adopted only the two main terms, such as in this example: "forest landscape restoration" OR "forest and landscape restoration." From the 31 initiatives, 19 returned documents matching our inclusion criteria (Supplementary Material 2). We selected documents containing information on at least one of these topics: FLR concepts, project requirements, characteristics, or principles. While reviewing these documents, we considered as principles "a fundamental law or rule as the basis for reasoning and action" ([46], page 20).

We excluded annual reports, texts for fundraising purposes and presentations (e.g., slides). When several documents were available from a single organization, we selected a maximum of three for each of them. We chose those that appeared first in Google search (43 documents; Supplementary Material 2), because this procedure returns the most relevant documents to the search. We also registered the location of the FLR initiatives described in the documents and the country of publication, which referred to the country of the organization or of the first author signing the article/document.

Overall, a total of 137 publications (94 scientific articles and 43 practitioners' documents) were retained in the sample without a quality appraisal (Supplementary Material 3). We hereafter referred to "papers" for scientific articles, "documents" for practitioners' documents, and "sources" for both types. Assessing document quality in qualitative systematic reviews is a controversial issue, although occasionally performed [40]. However, the procedure was inadequate in our study because our goal was to document the different perspectives across authors and initiatives rather than evaluate the quality of the evidence.

### 2.2. Coding and Textual Analysis

We followed Braun and Clarke [36] and Thomas and Harden [40] guidelines for coding and textual analysis, which was organized in three steps:

i.    We divided the central question we wanted to answer (i.e., which are the FLR principles commonly held by researchers and practitioners) into (i) concepts and definitions, and (ii) principles and criteria. We actively read a sample of the whole dataset before coding to gain familiarity with the material. This initial reading was active, paying attention to patterns, similarities, and meanings [36].

ii.   We proceeded to coding (i.e., identifying that a certain text is attributable to a relevant issue for the analysis [47]), and created a codebook (Supplementary Material 4), which represents a list of terms to label text portions (e.g., landscape definitions, intervention types) and document characteristics (e.g., year, author types). We read the text, highlighting important fragments and assigning them to one or more codes. Our approach to coding was a mix between top down and bottom up, because we departed from a list of aspects we wanted to investigate in the literature (e.g., FLR definition, principles, practices), but included themes as they appeared in our first reading of the literature. In addition, when inadequacies in the initial codebook were perceived, we included/excluded codes and changed their scope to a narrower/broader level [47]. All steps followed Thomas and Harden (2008) guidelines and relied on EPPI-Reviewer® v. 4.

iii.  After coding all documents, we analyzed text fragments looking for repetitions, similarities and differences [48]. With these procedures, we advanced new analytical themes from the information in the original literature, and generated concepts or understandings based on our judgment and ideas [40].

## 3. Results

### 3.1. Literature Characteristics

Except for a notable number of publications in 2005, the number of FLR publications steadily increased from 4 in 2010 to 21 in 2017, and more substantially since 2014 (Supplementary Material 5; summary of the WWF program in Mansourian and Vallauri [26]).

While FLR implementation is widespread among countries in all continents (being the obvious exceptions Antarctica and Artic), most of the publications came from developed countries in North America, Oceania and Europe (Figure 1).

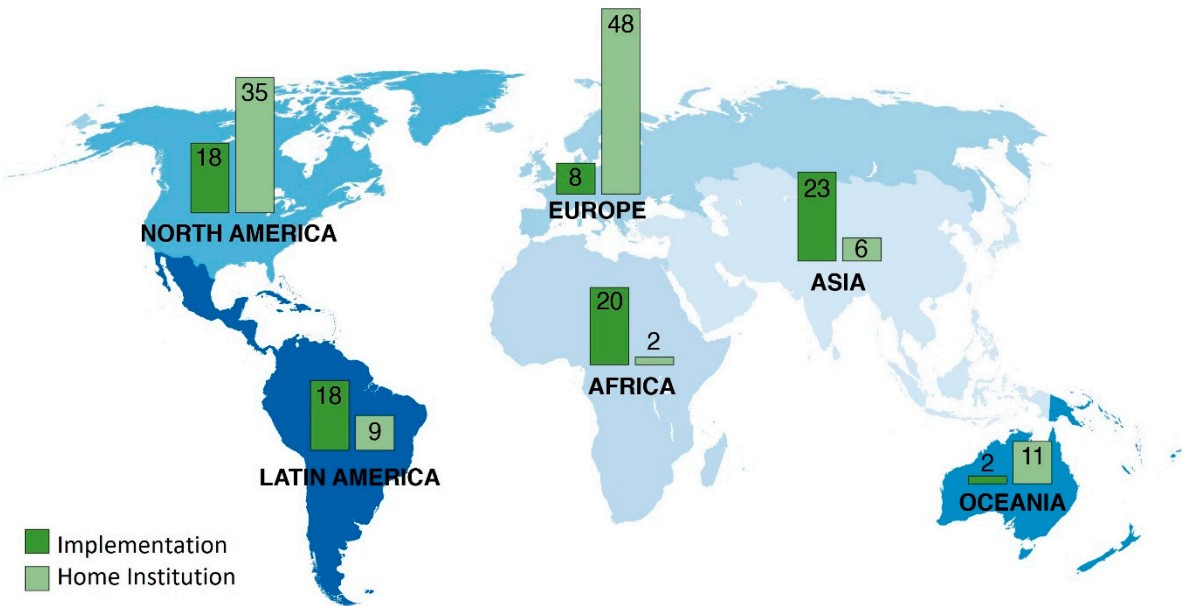

**Figure 1.** Forest and Landscape Restoration (FLR) sources surveyed by continent. "Implementation" refers to where FLR initiatives were established, while "Publication" refers to the country of the organization or of the first author of the source.

### 3.2. FLR Definitions and Aims

The terminology used in the published literature to describe FLR is diverse and has changed over time. Most sources (76.2%) refer to it as "Forest Landscape Restoration" although others adopt "Forest and Landscape Restoration" (16.6%) and "Forest Restoration in Landscapes" (1.4%). Sources adopting Forest and Landscape Restoration are more recent (2013–2017), denoting a change not only in terms of adoption, but also in concept, as the sector has broadened in scope beyond forestry and forest ecosystems. The most widely adopted FLR definition (38%), in both papers and documents, was proposed in 2000 by the WWF and IUCN (Table 1). It refers to FLR as "a process that aims to regain ecological integrity and enhance human well-being in deforested or degraded forest landscapes," [49] or minor variations in terms that do not generally affect its meaning, with a few exceptions, such as whether FLR is a planned process or not [50].

The concepts associated with the practical translations of FLR definitions during implementation vary in certain aspects, such as how the landscape is defined, how scale is incorporated (temporal and geographical), and how ecological dimensions are considered. Landscape definition varied across sources depending on disciplinary viewpoints [51,52]. However, authors agree the landscape is a heterogeneous mosaic of land uses, which may include old-growth and early successional forests, managed forests and non-forest lands, including agricultural and degraded lands [53,54]. Some authors emphasize the dynamism inherent to the landscape, classifying it as a human–environment interaction system [1,55].

Sources tended to agree on the importance of addressing geographical and temporal scales of FLR. At the geographical scale, the landscape was often generically defined as a continuous area, smaller than an ecoregion, but larger than a single site [56,57], which differs from its neighboring lands based on ecological and human aspects [56]. As for the temporal scale, authors concur that FLR is a long-term process [44,58], which aims to achieve a range of improvements in the ecological and human aspects of the landscape, through restoring forest functions, generating ES, and managing trade-offs between competing objectives [59,60].

While this information was not included in our codebook, we observed that sources usually do not specify who defines the landscape spatial and temporal scale. Boedhihartono and Sayer [30] describe FLR programs as a program of "seeking" solutions among stakeholders, rather than planning it top down. Including all stakeholders in the decision-making process from the early stages of FLR implementation contributes to project success and longevity, as indicated by recently published literature [8,13].

### *3.3. FLR and Associated Concepts*

Here we list the main concepts associated to FLR found in the literature.

### 3.3.1. FLR Benefits and Contributions

Although often based on *a priori* aspirations instead of demonstrated empirical outcomes, the sources mentioned the following expected positive outcomes: (i) ecological, (ii) economic, and (iii) social. FLR was also mentioned as contributing to achieve several harmonized international restoration goals [61,62] (Table 1).

### 3.3.2. FLR Planning and Implementation

Several sources reinforced the need to clarify FLR objectives in projects [63], and to understand the ecological, socio-economic and political contexts and available technical options [63] to achieve the desired outcomes [25]. Planning FLR must encompass short-, medium- and long-term activities [52]. Because restoration is a dynamic process, schedules need periodic revision [58]. The main FLR planning and implementation phases identified are listed below.

### Defining a Landscape

Sources suggest that FLR implementation should begin by identifying the landscape unit, and a few published guidelines were provided on this. The systematic approach entitled "Restoration Opportunities Assessment Methodology" (ROAM), developed jointly by the International Union for Conservation of Nature (IUCN) and the World Resources Institute (WRI), may contribute to assess the degradation types, and to identify priority areas and approaches for restoration in the landscape, but it does not provide specific guidelines to define the landscape unit [64,65]. Less systematized suggestions include the consideration of geographical and land-use characteristics through the use of maps, GIS, mathematical models, remote sensing inputs (e.g., aerial photos), and field data collection (e.g., ground-based observations) [58,66,67].

Although systematic protocols may help, the reviewed studies emphasize the impracticality of defining a single landscape scale applicable to all FLR projects [1]. As the biological and human aspects need to be addressed case-by-case, the landscape scale will likely vary across FLR initiatives in different socio-ecological contexts [63]. Yet, the lack of a clear technical approach to defining a landscape will result in a wide variation of landscape size and increases the challenges involved in comparing different initiatives and applying a single monitoring framework.

### Choice of FLR Interventions

A suite of FLR interventions was described in the sources, including a wide range of options varying from assisting natural regeneration to commercial tree plantations and other interventions to reduce degradation (e.g., [26] and Table 1). The choice of FLR intervention, its spatial extent and location are derived from project goals and context-dependent ecological and human features of the landscape, such as previous land uses, proximity to forest remnants, human population density, and distribution of settlements [68,69]. Natural regeneration was considered by certain authors as the most desirable solution because of its benefits, scalability, and lower cost when compared with tree planting [2,58]. Among tree-planting practices, agroforestry was highlighted as the approach with the highest

potential to generate human benefits. It allows expansion of tree cover while producing food [70] and generating other livelihood benefits, acting as a restoration "wild card".

The choice of the restoration strategies was directly linked to the location and conditions where those strategies were implemented [68,69]. Dudley and Vallauri [71] emphasized the importance of identifying where forests are needed, since FLR does not aim to restore forests across the entire landscape due to other land-use claims or ecological constraints. Thus, interventions must prioritize usefulness regarding socio-economic, political, and ecological perspectives [56,63,71].

Regarding prioritization, Orsi [72] presented guidelines for ranking sites in which forest restoration should be directed towards areas: (i) currently deforested, which were originally forest or woodland; (ii) with nearby existing forests; (iii) with large potential to conserve biodiversity, and (iv) sparsely human-populated. Locations with a mix of ownership and land tenure types were described as restoration challenges, when compared with landscapes dominated by few large properties. For example, Samsuri et al. [73] advocate that large/richer landowners may be less prone to participate on FLR initiatives in a watershed in Indonesia, while poorer people more commonly join restoration practices to increase their income levels. In areas densely populated and with major demands for food and forest products, the most suitable approach suggested is "mosaic restoration", which is a land-sharing strategy that integrates trees with existing land uses, such as smallholder cropping and grazing, resulting in multifunctional landscapes [58].

### 3.3.3. Monitoring and Adaptive Management

Monitoring must be based on FLR objectives, and its quality and cost efficiency depends on devising a minimum set of essential indicators at site and landscape scales, and predicting—as much as possible—actions at different timescales [74]. Examples of socio-economic, ecological, financial and overall project aspects to monitor include those described in the Collaborative Forest Landscape Restoration Program in the United States [65,74]. Such activities can be carried out not only by natural and social scientists, but also by local communities or locally trained personnel engaged in participatory monitoring [75]. Given the myriad FLR activities, their dynamic contexts and the long timeframe to achieve many restoration outcomes, monitoring must be kept flexible to allow for adaptive management through learning by doing and improving practices over time [20,75,76].

### 3.3.4. Socioeconomic Outcomes

Concepts referring to socioeconomic outcomes of FLR encompasses humanl well-being and human and institutional capital.

### Human Well-Being

Human well-being includes material and nonmaterial aspects, but the former is more often considered. Although seldom highlighted (Table 1), well-being improvements also come from nonmaterial benefits associated with FLR interventions, when landscape beauty, environmental quality, or recreational opportunities are enhanced [44,67] or when physical health, for instance, is impacted by increasing water potability on reducing natural hazards [58].

### Social and Human Capital

Project planning and implementation were commonly recommended to be participatory for four main reasons. First, including certain external partners (e.g., companies, private owners, research institutions, and NGOs) may allow technical improvement or addressing gaps in capacity or financing for implementing FLR [76,77]. For instance, partnerships with the public sector can be promoted by new legal frameworks that drive investments. Second, participation addresses the needs of local communities and less-influential stakeholders [45,63]. Thus, sources often argue for the importance of discussing stakeholders' objectives and needs through workshops, meetings, and other activities that

enable participation [78,79]. Third, participation is important whenever conflicts arise. For instance, Mansourian et al. [63] highlight that economic value shifts in the landscape under restoration might generate conflicts, such as from the misuse of natural resources or exacerbating inequalities. The implementation of conflict-resolution strategies, such as hiring mediators or facilitators, is therefore recommended to avoid/mitigate such problems [79]. Fourth, the participation of communities in projects increases human and social capital through enhancing leadership and other capabilities, and develops their potential to influence policies and improve self-esteem [44,67,75].

The success of FLR projects also depends upon building local human capital, through more access to scientific information, technical assistance, and capacity-building to restoration interventions [80]. Inclusive processes should also go beyond sharing scientific and technical knowledge with local people, to incorporate their traditional knowledge on restoration strategies [60]. Certain authors argue that, to guarantee restoration success, forest agents should focus especially on small landowners and marginalized communities responsible for restoration implementation [44,81–83]. In addition to enhancing project success rates, education, training and capacity-building may increase job and income opportunities beyond the project itself.

### Institutional Capital

Institutional capital (i.e., informal and formal rules, such as laws and policies) drives land-use decisions [44,70,82]. For effective compliance with legal instruments, organizations leading FLR implementation should assist local processes by promoting an adequate governance structure, strengthening the capacity of public institutions, engaging the private sector and markets, encouraging the equitable participation of stakeholders and, consequently, decentralizing decision-making to local groups [45].

### 3.3.5. Ecological Outcomes

Ecological outcomes encompasses the concepts directly related to biodiversity, ecological processes and ecosystem services.

### Biodiversity Conservation

In highly fragmented and degraded landscapes, FLR can address a long-term solution for improving ecological functionality and agricultural productivity [75] by reducing pressure on natural forest remnants, augmenting their buffer zones and improving landscape connectivity [54,70,75,84]. Planting native species (e.g., in agroforestry, enrichment or mixed-species plantings) is recommended for ecosystem restoration and genetic diversity conservation [25,85,86]. The presence of seed sources (e.g., forest remnants and populations of targeted species) in the landscape ensures the availability of propagules for seedling production and to foment spontaneous regeneration in restoration sites through seed dispersal from remnants [85,86]. Supporting a network of seed collectors and high-quality seedling producers was also highlighted as a key aspect of restoration success [58,68].

Because not all species are able to colonize or persist in degraded or early successional forests, the protection of old-growth remnants was mentioned as crucial to conserve threatened species [85,86]. The control of superabundant and invasive species, protection against unwanted animals (e.g., uncontrolled grazing livestock and other ruminants), and enrichment of secondary forests [54,87] are important complementary actions that preserve local ecological functions in mosaic landscapes.

Examples such as adopting some non-native species, especially in agroforestry systems and monoculture tree plantations, show remarkable potential to contribute to the overall goals of FLR programs, with benefits for carbon sequestration, soil protection, commercial production, and water infiltration [78,88]. However, sources argue for the crucial role of balancing where, when and which species to use to prevent the wholesale conversion of native forests to commercial plantations that may lead to a cryptic loss of carbon stocks, biodiversity, and ES [25].

Climate Change Mitigation and Adaptation

The global urgency and emerging interest to mitigate climate change, exemplified in recent ambitious global agreements, is highlighted as an opportunity for advancing FLR initiatives [25,75,85,89]. Forest and Landscape Restoration interventions may alleviate climate change effects on biodiversity and ES provision at the landscape, such as establishing protected areas for watershed and nature conservation, promoting forest restoration, establishing buffer zones [54], and controlling fires [65]. In this context, FLR could replace degraded lands with sustainable land use based on landscape-management practices [1,44,83].

Increase the Provision of Ecosystem Services

One motivation for restoring degraded lands is to improve the supply of goods and services from ecosystems other than climate change mitigation and adaptation [83]. At the landscape scale, balancing different ES to minimize trade-offs amongst them is key to FLR success [90]. Payment for Environmental Services (PES) is mentioned as a tool to foment large-scale restoration, together with law enforcement, securing political and public will, and providing financial support [44,54]. In certain cases, PES is based on cost-benefit analyses, which may be based on estimating individuals' willingness to pay for restoration and its benefits, or land opportunity costs, for example. Chadourne et al. [67] highlight that a limitation of "cost-benefit analysis" is that restoration returns may be underestimated by the community, since some "direct-use values" for forests (e.g., recreational use and aesthetic value), and the "non-use values" (e.g., enhanced biodiversity, the existence values of plant and animal species, values associated with a unique culture embodied by their natural heritage) are difficult to identify and incorporate into PES schemes.

### 3.3.6. Landscape Multifunctionality

Landscape multifunctionality refers to synergies and complementarities in a landscape with multiple land uses, each one valued differently by individual stakeholders [89]. Applying landscape multifunctionality concepts in FLR improves the coexistence of different land uses, accomplishing a range of stakeholders' interests [91]. Analogous to results from ES studies, landscape multifunctionality entails different spatial patterns, trade-offs and synergies [89].

The integrative effort to restore multiple functions on a landscape, by creating a "mosaic" where protected areas, forest types, management interests, and various land uses are combined and connected [92], is one of the major differences between site-centered ecological restoration and the landscape approach of FLR. Any FLR project will compose a set of site-based interventions whose combination and integration provides significant landscape-level outcomes [93]. Because of this landscape-scale integration, identifying degraded land cover through multi-stakeholder consultations and reviewing relevant land use/cover maps and statistics are essential [54,89,94].

### 3.4. Guiding Principles of Forest and Landscape Restoration

Based on the 137 reviewed sources, we identified 12 FLR Principles (Table 2). We divided the principles into three domains: Project Management and Governance, Human, and Ecological (further detailed below) and listed key criteria that encompass each principle (Figures 2–4). In certain cases, a single criterion may contain more than one principle; for example, implementation of forest-based production systems may address Economic Diversification (Principle 8), Biodiversity Conservation (10), and Provision of Ecosystem Goods and Services (11). Such overlapping is expected given the multifunctional approach of FLR.

**Table 1.** Some of the main Forest and Landscape Restoration (FLR) themes addressed by 137 sources sampled in the literature review.

| Theme | Subject | Description | Sources Number (%) | Examples |
|---|---|---|---|---|
| **Definitions, aims and associated concepts** | FLR definition | "a process that aims to regain ecological integrity and enhance human well-being in deforested or degraded forest landscapes" | 52 (38%) | [49] |
| | Ecological integrity | "ecosystem composition, structure, and functional processes" | 20 (15%) | [26] |
| | Ecological functionality | "the goods, services and ecological processes that forests can provide at the broader landscape level, as opposed to solely promoting increased tree cover at a particular location" | 35 (26%) | [93] |
| | Return to pre-degradation conditions | Focus on recovering and enhancing ecological attributes across a landscape instead of returning to historical conditions and land use patterns | 23 (17%) | [95] |
| **Benefits and contributions** | Ecological outcomes | Generate carbon sequestration, biodiversity conservation, and water and/or soil protection | 42 (31%) | [63,75,82] |
| | Economic outcomes | Enhance land productivity, food security, create jobs and sources of income, such as trading forest products | 20 (15%) | [44] |
| | Social outcomes | Reduce inequality in food provision and reduce conflicts | 35 (26%) | [14,80] |
| | International restoration goals | Contribute to initiatives such as the Bonn Challenge, New York Declaration on Forests, Aichi Target 15 of the Convention of Biological Diversity | 62 (45%) | [61,62] |
| **Planning and implementation** | Restoration plantings | Plant trees for forest restoration in areas with low potential for natural regeneration or for commercial uses. Includes several plantings methods (*nuclei*, agroforestry, enrichment plantings, among others) | 35 (26%) | [54] |
| | Sustainable practices in agricultural lands | Control erosion, reduce environmental disturbances, diversify crops and techniques, improve supply of ES | 26 (19%) | |
| | Prioritization of activities | Develop guidelines for site selection and prioritization of restoration. | 25 (18%) | [72] |
| | Monitoring and adaptive management | Apply indicators for monitoring human (e.g., well-being, income, employment, food security) and ecological (e.g., forest structure, function, composition and connectivity) outcomes | 26 (19%) | [44,96] |
| **Human outcomes** | Non-monetary benefits | Improve human well-being by enhancing livelihood resilience, physical health or recreational opportunities (due to improvement of landscape beauty) | 9 (7%) | [44,67] |
| | Gender equality | Improve gender equality and provide opportunities to marginalized groups. | 7 (5%) | [27,60,83] |
| | Legal compliance | Boost compliance and enforcement of the law in the project landscape | 12 (9%) | [14,63] |
| **Ecological outcomes** | Biodiversity conservation | Reduce pressure on natural remnants, such as augmenting buffer zones, reducing degradation and deforestation | 23 (17%) | [54,70,75,84] |
| | Environmental services | Increase supply of ES including water flow regulation (17%, of sources), carbon storage (13%), but also ecotourism (6%) and pest control (2%), for example. | 38 (28%) | [90] |

**Table 2.** Project management, Human and Ecological Principles for forest and landscape restoration based on 137 reviewed sources.

| Domain | Principle | Description |
|---|---|---|
| Project management and governance | 1. Landscape scale | FLR initiatives should encompass an area larger than a single site/property but smaller than an ecoregion while encompassing multiple and integrated land uses. A case-by-case selection is necessary to adapt to the region and context where the FLR program will be implemented. |
| | 2. Prioritization | The choices and priorities regarding the type, location and timing of restoration and conservation interventions should be planned to enhance positive ecological and human outcomes, based on human, social, economic, political and ecological perspectives. |
| | 3. Legal and Normative Compliance | FLR programs should be planned and implemented in order to comply with every applicable legislation but also follow local/ traditional norms. Organizations leading FLR implementation should promote adequate governance structures, strengthening capacities, and implementing strategies that avoid conflicts. |
| | 4. Participation | FLR programs should aim to achieve full collaboration (co-design and co-manage), by promoting the active participation of local communities and other stakeholders to achieve shared responsibilities in planning and decision-making. The demands and needs of all stakeholders should be incorporated to avoid conflicts and to better manage the likely trade-offs between ecological and human outcomes. |
| | 5. Adaptive Management | FLR activities and outcomes regarding ecological and human aspects should be monitored throughout the initiative. Knowledge generated from monitoring allows reflection and adaptive change in project activities. |
| Human Aspects * | 6. Enhance Livelihoods | FLR programs should seek the improvement of local livelihoods through the delivery of financial outcomes, poverty alleviation, food security, and non-material benefits such as landscape beauty and recreational opportunities. Activities planning and implementation considering the improvement of local livelihoods in economic and non-economic terms (i.e., human and social capital). |
| | 7. Inclusiveness and Equity | FLR should aim to include and benefit all stakeholders, paying a special attention to gender equality and less powerful subgroups in society. |
| | 8. Economic Diversification | Needed to achieve more resilient livelihoods and also to coordinate land uses across the landscape to achieve ecological outcomes. The practice of FLR should seek to integrate diverse land uses and economic activities in the selected landscape. |
| | 9. Capacity Building | FLR programs should provide local stakeholders with relevant scientific and technical information and assist in building capacities of individuals and groups to participate fully in FLR activities. Local and indigenous knowledge about regeneration, restoration and land management practices should inform co-designed FLR programs. |

**Table 2.** *Cont.*

| Domain | Principle | Description |
|---|---|---|
| Ecological Aspects | 10. Biodiversity Conservation | FLR programs should protect and conserve native ecosystems, including both forest and non-forest ecosystems, such as savannas and grasslands. Native species should be preferred over exotic species, while also representing a diversity of functional groups and enhancing genetic variability. |
| | 11. Landscape Heterogeneity and Connectivity | FLR programs should improve landscape functionality by reducing pressure on natural forests, fostering agricultural practices that enhance the permeability of agricultural lands to the native fauna and flora, improving forest connectivity and fomenting positive land-use synergies at the landscape level. |
| | 12. Provision of Ecosystem Goods and Services | FLR programs should encompass ES provision and therefore carbon stocks, natural pest control, pollination services, ecotourism, and the provision of water, food, timber, and non-timber forest products. Interventions should also aim to improve soil retention, and to reduce soil erosion and sedimentation. Programs should aim to minimize the trade-offs among ES. |

* Includes not only social and economic terms, but also attributes at the individual level (e.g., human capital). Although human dimensions are often related to the literature on psychological determinants of human behavior, we chose this term instead of "human and socio-economic dimensions" to keep the term more succinct.

### 3.4.1. Project Management and Governance Principles

We identified five principles for planning and governing FLR projects as a result of the literature review (Figure 2). The first principle, i.e., "Landscape Scale", incorporates techniques and methods that may contribute to identify the appropriate landscape areas for FLR programs, as well as the ecological and human characteristics that need consideration when advancing this choice. Because landscape definitions vary across studies, we found few consensual "rules" on how to select and delimitate landscape areas. Mansourian et al. [63] proposed that landscape delimitation decisions must occur case-by-case. The "Prioritization and Optimization" principle originated from coding the several types of restoration and conservation intervens that FLR initiatives may adopt. The literature agrees that a range of strategies and interventions should be adopted in FLR initiatives to achieve better ecological and human outcomes and to foster multifunctional landscapes.

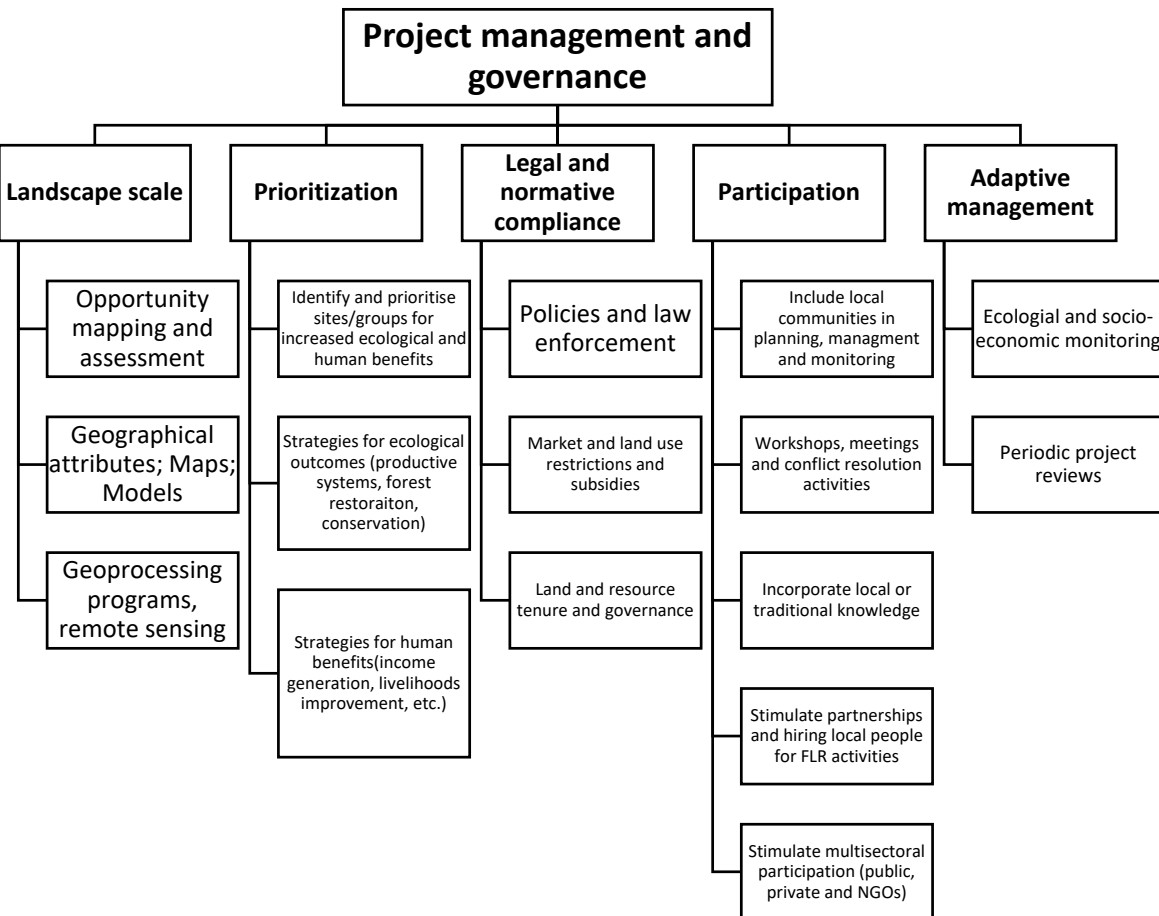

**Figure 2.** Management and governance principles (top boxes) and criteria (lower, branched boxes) of FLR based on the review of 137 sources.

The third principle, i.e., "Legal and Normative Compliance", means that FLR programs should follow all the applicable legislation, including municipal, state, and national legal instruments. Additionally, knowledge about informal norms, such as traditional rules regarding land or resource uses and activities, should be considered in order to increase the likelihood of FLR success and to minimize conflicts, since these norms are often more important than legal ones to explain resource use [78]. Well-designed and decentralized governance structures, integrating all stakeholders and processes, have been considered crucial to advance not only compliance with legal instruments and traditional norms, but also to supervise programs/activities along FLR phases and to adapt management practices.

The "Participation" principle should take place in every stage of FLR, from planning to monitoring. Although FLR programs, depending on the local context, may need to begin with more consultation-type participation, the success and sustainability of programs will likely depend on program co-design, co-management and co-monitoring with applicable governance.

The "Adaptive Management" principle refers to the use of an experimental approach in which FLR interventions are monitored and assessed to determine whether each strategy achieved the desired outcomes [97], and then strategies are adapted accordingly. Monitoring and assessment are central to this issue, as are the flexibility in the approach to adapt to increased knowledge on the issue.

### 3.4.2. Human Aspects

The reviewed information on human dimensions (Section 3.3.4) was consolidated into four principles (Figure 3). The "Enhance Livelihoods" principle refers to the aim of enhancing local well-being indicators, therefore encompassing the economic capital (monetary and nonmonetary income, food security, and poverty alleviation) and nonmaterial benefits such as landscape beauty and recreational opportunities. Enhancing livelihoods was cited in only 15% of sources, although cases refer more frequently to monetary income than to the use of local extracted or produced resources, which can contribute to food security, or nonmaterial benefits. Actions needed to achieve well-being improvement tend are poorly detailed.

The "Inclusiveness and Equity" principle emphasizes equity in terms of access to FLR benefits, but also to avoid burdens resulting from restoration. The burdens of ES programs are difficult to assess because (i) studies focus on a delimited landscape measuring the aggregated benefits rather than impacts, (ii) the impact of ES programs is spread through space and time, and (iii) there are still knowledge gaps in cross-scale and cross-location impacts. These topics favor unaccounted externalities that may reduce the global benefits of ES programs [98]. Gender equality is more often highlighted, but other vulnerable groups are also cited [44].

"Economic Diversification" addresses the need to diversify activities, strategies and land uses across the landscape to achieve the expected human and ecological outcomes. This means that FLR programs should focus not only on forest restoration and tree planting but also on a diversity of integrated land uses and economic activities to achieve more resilient landscapes [25]. It also includes different sources of income (carbon market, PES, public and private funds) that could support FLR activities [99]. The activities of FLR also have the potential to generate jobs, income and capacitation throughout the restoration supply chain, which includes nurseries, suppliers of equipment and fertilizers, and the planting and monitoring teams. For example, Banks-Leite et al. [100] estimated that $198 million is required annually for three years to restore the Brazilian Atlantic Forest to 30% of its original cover, which would conserve most of the species of this domain and significantly foment the restoration supply chain.

As a means to achieve FLR, but also as an end in itself, the "Capacity Building" principle refers to building human capital, most importantly regarding people's knowledge (scientific, technical, traditional) and capacities to participate and codesign/manage FLR activities and restoration supply chain activities.

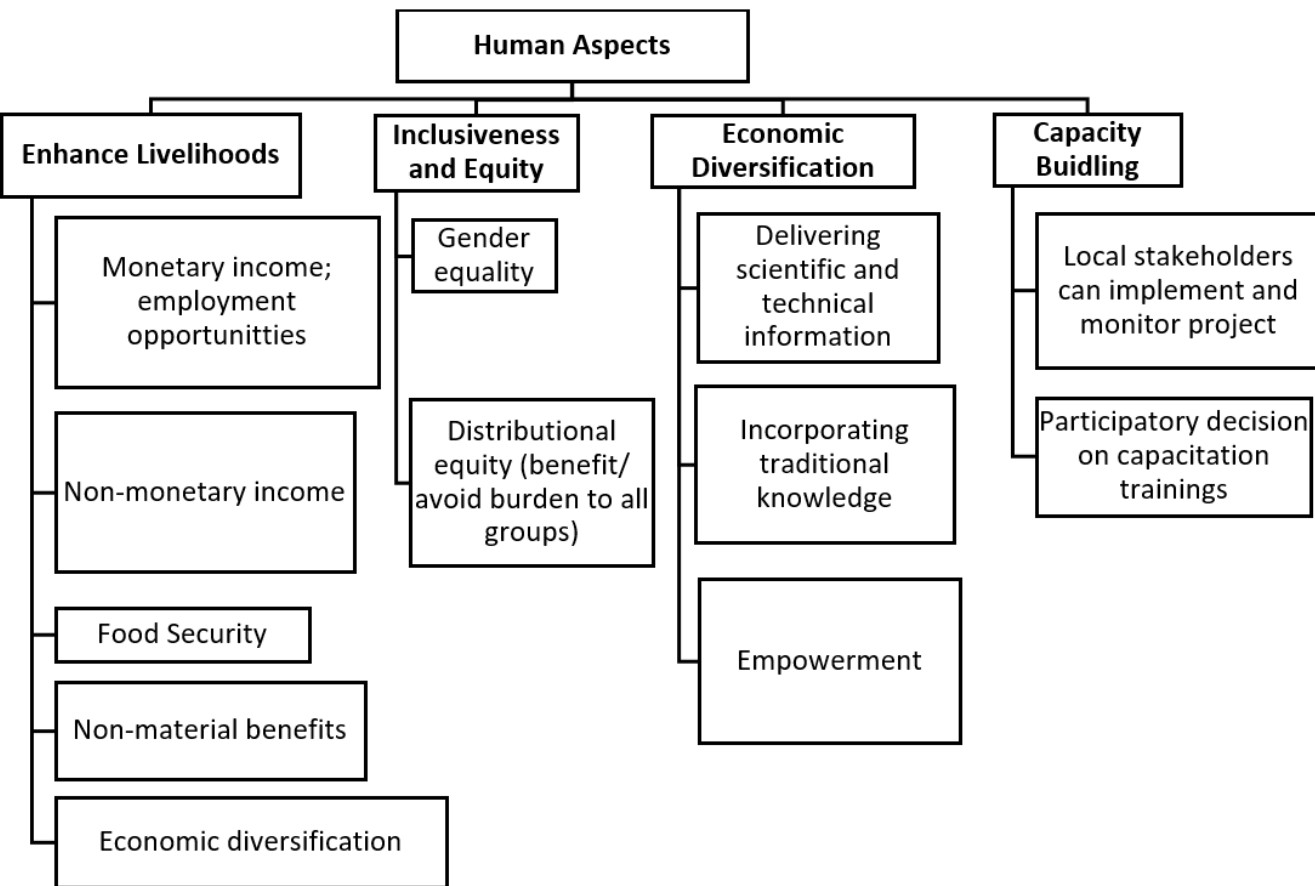

**Figure 3.** Human principles (top boxes) and criteria (lower, branched boxes) of FLR based on the review of 137 sources.

3.4.3. Ecological Aspects

The analysis of codes related to the Ecological domain of FLR resulted in three principles (Figure 4). "Biodiversity Conservation" addresses the indispensable protection of native habitats and species, as well as the use of forest restoration as a strategy to enhance biodiversity conservation. Interventions aimed at restoring forest remnants also emerged as an aim, such as the control of superabundant and invasive species (e.g., grazing livestock and other ruminants) and enrichment plantings [54,87]. "Landscape Heterogeneity and Connectivity" considers the assessment of landscape configuration and changes, directly linked to monitoring forest connectivity, improvement in forest cover area, diversification and heterogeneity of land uses, and increase in the permeability of agricultural lands to native species flow. Landscape heterogeneity (i.e., the different land covers in the landscape) interacts with the third Ecological principle: "Provision of Ecosystem Goods and Services". For example, native remnants and tree plantings could foment biodiversity conservation and ES provision. At the same time, the planned implementation of different production systems (from agroforestry to pastures and more intensive production systems, such as sugarcane) could attend to the economic needs of different groups of farmers and private companies, which also directly benefit from some of the ES generated at the landscape scale. Planning the location of land uses in the landscape with local stakeholders is also key for FLR projects [30]. Additionally, legal mechanisms may also allocate native forests for the restoration of more environmentally sensitive areas [101].

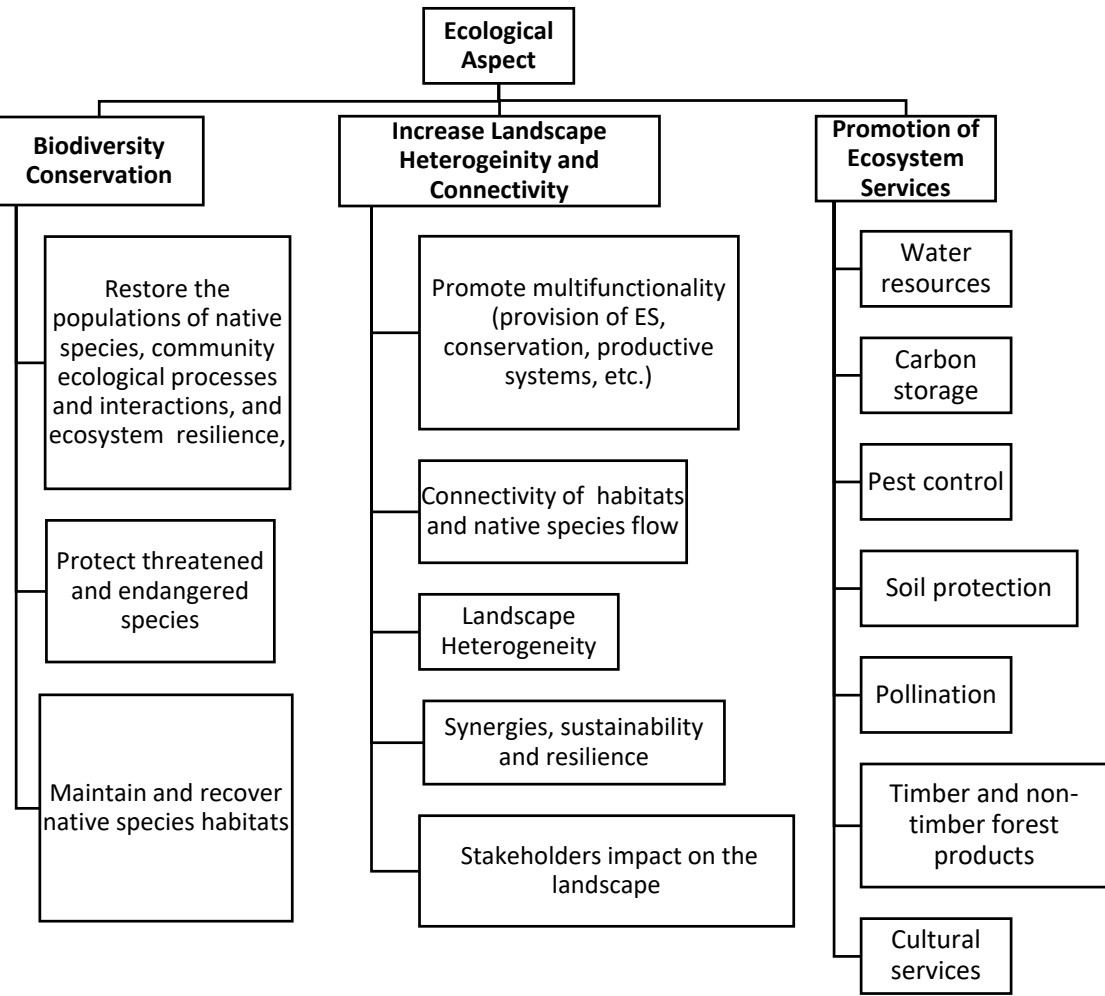

**Figure 4.** Ecological principles (top boxes) and criteria (lower, branched boxes) of FLR based on the review of 137 sources.

## 4. Gaps and Recommendations

While we provide an overview of the FLR literature, the principles that percolate it have already been raised by other authors. The six FLR principles defined by Besseau et al. [24] encompass those identified in the sources sampled here. Although these authors do not explicitly mention "Legal and Normative Compliance" and "Prioritization" principles, these principles are contained implicitly. On the other hand, the aspects of literature on principles "Enhance Livelihoods" and "Economic Diversification" are also captured in the principle "Inclusiveness and Equity", which include health and economy. Besseau et al. [24] also does not provide a literature review or go deeply into the principles. indicating that the core FLR principles defined by GPFLR reflect the literature on this subject, although adoption of principles by the literature is heterogeneous. For example, Besseau et al. [24] address the principle of local context and long-term adaptive management, key FLR aspects that are scarcely mentioned in the sources sampled but are key for FLR success [32].

Brancalion and Chazdon [25] raised valuable human and ecological principles for FLR initiatives, pointing out common pitfalls in forest restoration and quantification of ecological benefits while highlighting improvement of well-being and landscape heterogeneity for achieving multiple livelihood outcomes targeted by FLR programs. Djenontin et al. [102] present a thorough assessment of factors related to governance and project management in FLR from the household to the regional scale. We add to their work by exposing Project Management and Governance Principles such as Restoration Prioritization and, to some

degree, Adaptive Management that have not been explicitly listed although they are well known by practitioners.

Project results must be monitored to quantify the benefits and identify both successful strategies and aspects for improvement of FLR initiatives. Monitoring is highlighted for the activities encompassed in FLR, such as forest restoration and income, and more recent publications highlight this need and provide guidelines ([103] and references in [104]). Nonetheless, part of the literature sampled and reviewed here does not reinforce the need to use landscape-level monitoring indicators. Reed et al. [104] mentioned that, while most study cases of integrated landscape management claim to be successful, only 6% of the study cases in the grey literature provided robust evidence of success. The difficulty and bias in FLR evaluation may be caused by (i) varying perceptions of what is a successful outcome among stakeholders, (ii) lack of a common standard of evidence for success, (iii) demotivation to report on project failures, (iv) lack of resources for monitoring and evaluation, (v) short-term funding (2–3 years) of projects [104].

Another point is that the reviewed documents overlook the need to evaluate people who have not actively participated in the FLR program but are nevertheless impacted by FLR interventions. Only a few sources addressed how to measure and monitor ES within FLR programs. While PES and market incentives can be a powerful tool to foment forest restoration [105], provision of ES is dynamic between land uses and over time, presenting trade-offs between services and landowners' preferences [106]. We argue that the lack of careful monitoring of the FLR benefits derived from ES provision may be a bottleneck for developing PES programs.

The literature on the principles of FLR agrees on the benefits to people as key components for FLR programs (e.g., participatory processes, well-being improvement, provision of material, and nonmaterial goods and services). Similarly, improving local livelihoods and economic outcomes are also key for FLR project sustainability and stakeholder engagement and empowerment. Nevertheless, we found few guidelines for developing such activities in the review, especially regarding the distribution of benefits and the reduction of external burdens that could impact other areas. Due to the challenging aspect of equitable distribution of FLR benefits in the long term, this aspect must be incorporated by FLR programs in order to overcome it [98]. While tailored frameworks are necessary [31]—given the unique context of each landscape—guidelines can be established by FLR initiatives for greater participation of local stakeholders, thus reducing the burden on coordinators and increasing the chance of success [13,30,107]. Practical recommendations for developing such activities could greatly contribute to future FLR initiatives, as each landscape requires unique and dynamic socio-ecological strategies. On the other hand, guidelines may be difficult to establish beforehand and may not be universally relevant.

An important gap is evidenced by the relative scarcity of literature addressing the human aspects (socio-economic and otherwise) of FLR, despite their centrality to the concept. Because FLR was born within the natural sciences, this gap was expected. Robinson [108] observed that most practitioners and researchers recognize the importance of human aspects in restoration but are unable to incorporate them due to the lack of time, funds or technical expertise. Human aspects are still more frequently addressed as a means to achieve restoration than as an end in themselves, which is worrisome because continuous stakeholder engagement since the early stages of FLR is key for long-term success [8].

Above all, improving people's economic outcomes from FLR involvement is advised [105], which assumes that economic benefits are the main drivers of people's behavior and that people are able to compute and optimize outcomes across alternatives. Instead, there is a large literature showing that humans are boundedly rational—i.e., unable to process large amounts of information [109]—and may decide instead based on shortcuts such as heuristics (e.g., availability heuristics; [110]), or influenced by the social context and social norms [111]. Moreover, principles are often suggested in the literature without a foundation on empirical evidence from FLR projects, and consequently do not consider landscape attributes, which are even more central to the approach. While a few ecological

aspects, such as connectivity, are frequently cited as an FLR aim, there is little consideration of impacts on human, social, or cultural capital at the landscape scale, except perhaps for economic diversification. For instance, if we take the ecological literature as a baseline, is the richness of economic activities or its diversity (against dominance) a more important attribute to achieve more sustainable landscapes? And which are the human aspects more important to integrate land uses across the landscape? Restoration programs and FLR are usually coordinated by specialists in environmental sciences, who may overlook or misinterpret social aspects of their practice and research [112]. Given this gap and the importance of long-term social benefits of FLR, we stress the importance of multidisciplinary teams in implementing and monitoring FLR initiatives.

## 5. Conclusions

FLR is a promising approach to generate multiple benefits and tackle some of the most pressing environmental challenges of the Anthropocene. Ecological principles are well-recognized within FLR programs, and despite the under-representation of human aspects in the scientific literature on restoration, these aspects were more often included in the "grey literature" of FLR initiatives. FLR has evolved to achieve integration of ecological and social objectives. Our results help to fulfill a knowledge gap in restoration science while also serving as a starting point for developing new tools, guidelines, frameworks, standards, and accountability schemes that could greatly improve FLR effectiveness, avoid unintended consequences, and increase transparency.

**Supplementary Materials:** The following are available online at https://www.mdpi.com/2073-445X/10/1/28/s1.

**Author Contributions:** Conceptualization, L.B., C.G.B., C.M., R.A.G.V., R.L.C., V.G., and P.H.S.B.; methodology, L.B., C.G.B., and C.M.; software, L.B. and C.G.B.; validation, L.B., C.G.B., and C.M.; formal analysis, L.B., C.G.B., and C.M.; investigation, L.B., C.G.B., and C.M.; resources, L.B., C.G.B., C.M., V.G., and P.H.S.B.; data curation, L.B. and C.G.B.; writing—original draft preparation, L.B., C.G.B., C.M., R.A.G.V., and P.H.S.B.; writing—review and editing, R.G.C., C.M., R.A.G.V., R.L.C., V.G., and P.H.S.B.; supervision, C.M., R.A.G.V., V.G., and P.H.S.B.; project administration, C.M., R.A.G.V., and P.H.S.B.; funding acquisition, V.G. and P.H.S.B. All authors have read and agreed to the published version of the manuscript.

**Funding:** This research was funded by WeForest asbl. C.B. and L.B. received funding from Higher Education Improvement Coordination (CAPES), Processes # 88882.180110/2018-01 and 1708853 respectively.

**Institutional Review Board Statement:** Not applicable.

**Informed Consent Statement:** Not applicable.

**Data Availability Statement:** The data presented in this study are available on request from the corresponding author.

**Acknowledgments:** The authors would like to thank three anonymous reviewers who provided valuable comments on the earlier versions of this manuscript.

**Conflicts of Interest:** The authors declare no conflict of interest.

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
