# Peer review of "Forest and Landscape Restoration: A Review Emphasizing Principles, Concepts, and Practices"

_land, doi:10.3390/land10010028_

Round 1

Reviewer 1 Report

The paper contributes to a better understanding on some key FLR-related concepts and definitions adopted in the English literature. Its main contribution is, however, in providing a more comprehensive formulation of guiding principles of FLR based on the review.

The review has apparently focused on English literature. If so, the authors should state that and acknowledge this as a limitation of the study. FLR-related terms and concepts tend to vary from country to country, sometimes complicated by the way some of the common terms in the English language are translated in the country’s language.

Considering the paper’s objective related to the identification of “the underlying strategies commonly suggested to enable FLR implementation in different socio-ecological contexts”, the results for some of the attributes searched are described too superficially or partially. That’s particularly the case for 3.4.2 – Choice of FLR interventions.  

The economic dimension is not sufficiently covered in the guiding Principles. The description given for the principle “Economic diversification” does not include a criterion directly addressing the economics of restoration interventions, including value-added chains from restoration products or services.

Section 5 on Gaps and Recommendations is weak. I didn’t find specific recommendations in relation to how the principles & criteria could be best used/internalized in FLR initiatives at various scales (national to local).  

SPECIFIC COMMENTS

Item 2.1, Terms searched (line 102): The term “recovery” could have been included as it is often used in some countries (for instance in Latin America).

Item 2.1, “publications documents”: ?

Item 2.1, “Grey” literature (line 114): Technical reports and conference/workshop proceedings are important sources in this category.

Item 2.2, Table 1 (line 165): The question “Is heterogeneity part of what guides FLR” is a too generic question. It should be more specifically framed.

Item 3.2, “in term adoption” (line 182): It should be “in terms of adoption”.

Reviewer 2 Report

The first two references are not numbered, nor is IUCN in line 47.

Line 61: I would add the reference McLain et al. 2918 (see below).

Line 113: you are missing several references on FLR that are available on Scopus.

The list of documents reviewed needs to be in alphabetical order.

One particularly relevant article (that looked at discourses in FLR) is that of Reinecke and Blum (Reinecke, S. and M. Blum. 2018. Discourses across scales on forest landscape restoration. Sustainability 10:613.)

But there are several other key ones that are not included, e.g,:

Djenontin et al. 2020 (Ultimately, What is Forest Landscape Restoration in Practice? Embodiments in Sub-Saharan Africa and Implications for Future Design. Environmental Management),

Dudley et al. 2018 (Measuring progress in status of land under forest landscape restoration using abiotic and biotic indicators)

Dudley and Maginnis 2018 (A stepwise approach to increasing ecological complexity in forest landscape restoration),

Mansourian 2018 (In the eye of the beholder: Reconciling interpretations of forest landscape restoration, Land Degradation and Development)

Mansourian and Parrotta, 2018 (The need for integrated approaches to forest landscape restoration, book published by Earthscan);

McLain et al 2018 (Toward a tenure-responsive approach to forest landscape restoration: A proposed tenure diagnostic for assessing restoration opportunities);  etc…

Line 117: you refer to 31 organizations, yet I only see 19 in the annex (and some are not working on FLR – e.g. SER – while others are that are not included in your list). For example, WWF has produced several documents in the last 20 years on forest landscape restoration that do not appear at all in your results, yet you refer to them in line 168? I just checked and a search on WWF International’s website for “forest landscape restoration” yields 705 results!

I find it difficult to see the link between the central questions in Table 1 and the results section (especially sections 3.2-3.8). It would be helpful to better mirror them.

The “Human aspects” dimension for the principles is a vague term. Furthermore, it can be argued that many of the principles in the preceding category are human. A better distinction is required here.

Since you note in the “gaps and recommendations” that your principles are captured in those of Beseau et al 2018, the reader is left wondering what you add to the debate.? it might be worth starting with those and demonstrating that your approach is a way of testing the validity of these principles?

Line 192, add the reference Mansourian, 2018 see above.

Line 527 this sentence is unclear.

Reviewer 3 Report

Please see my comments in the attached file. 

Round 2

Reviewer 2 Report

If I use your methodology for the search (which is the point of a "systematic review") I obtain a number of articles that you do not cite (or have excluded, possibly, even though they are about forest landscape restoration?) There is a significant lack of rigor in your application of your own methods.

Author Response

Response to Reviewer 2 – Manuscript ID land-1002264

Forest and landscape restoration: A review emphasizing principles, concepts, and practices

Comments provided by the Editor

By definition a “systematic review” includes all findings and not a
biased selection. As noted before at least one major organization is
missing in the review, and with it all of its tools and references. In
addition, a value of a “systematic review” is that it can be repeated.
If I carry out the search according to the methodology provided in the
article, I obtain different results. Several important articles are
missing (even some by some of the co-authors themselves, e.g. Brancalion).

We screened for sources that did not refer directly to FLR, as stated in lines 132-135: “We then screened titles and abstracts of publications based on their consideration of FLR and social and/or ecological aspects of restoration, resulting in 118 scientific papers. We then fully read these papers and, based on the same inclusion criteria, we ended up with 94 publications (Appendix 1).” It could be that the articles found by the reviewer may have been removed at this stage of data preparation.

If possible, we would like the reviewer to share the missing articles that he found in his search, so that we could look into it and i) justify why they are not included, or ii) recognize the missing articles.

In addition, there are errors in the annex: GRN is a project of the
Society for Ecological Restoration not an institution per se. As such,
the link is wrong and it should not appear as an “organization” in the
annex. “Biodiversity International” does not exist, the organization is
called “Bioversity International”. GPFLR is a partnership not an
organization.

We are thankful to the reviewer for pointing that out. Since these initiatives and organizations were suggested by practitioners of FLR and were included in the analyses, we corrected that and stated that we gathered the grey literature from ‘FLR initiatives and organizations’ in Supplementary File 2 and lines 140-144 of the Methods section. We also replaced the term ‘organization’ by ‘initiatives’ throughout the manuscript.

Biodiversity International was corrected to ‘Bioversity International’ in the Supplementary File 2, as suggested by the reviewer.

Reviewer 2

If I use your methodology for the search (which is the point of a "systematic review") I obtain a number of articles that you do not cite (or have excluded, possibly, even though they are about forest landscape restoration?) There is a significant lack of rigor in your application of your own methods.

We followed a standardized methodology to conduct our review and described the specifics as clearly as possible. As mentioned in the other comment to the reviewer above, we screened for sources that did not refer directly to FLR, as stated in lines 132-135: “We then screened titles and abstracts of publications based on their consideration of FLR and social and/or ecological aspects of restoration, resulting in 118 scientific papers. We then fully read these papers and, based on the same inclusion criteria, we ended up with 94 publications (Appendix 1).” It could be that the articles found by the reviewer may have been removed at this stage of data preparation. One issue is that not all papers use the terminology of FLR that could potentially be included in the review.

If possible, we would like the reviewer to share the missing articles that he found in his search, so that we could look into it and i) justify why they are not included, or ii) recognize the missing articles.

Reviewer 3 Report

Please find attached my new comments.

Author Response

Response to Reviewer 3 – Manuscript ID land-1002264

Forest and landscape restoration: A review emphasizing principles, concepts, and practices

  1. Introduction:

The authors rewrote the last paragraph, with some clarity on what their objective, methods, and findings are. First, I suggest that the authors conduct some proofreading, especially focusing on the tenses of the sentences.

We conducted a throughout English review of the manuscript and tracked changes on several small changes. We hope that the English grammar is acceptable now.

Second, it would be good to elaborate the findings a bit more. Currently, the findings read: “More specifically, we identified the main FLR principles in the literature, identify gaps and provide recommendations based on existing established principles.” Here is one suggestion about what I meant in my comment regarding guiding the reader on the major results: “Our analysis uncovered some main FLR principles in the literature that we organized across three domains, including xxxxx. We also identified gaps such as xxxxxx and provided recommendations based on existing established principles.”

According to the reviewer’s suggestion, we added: Our analysis uncovered some main FLR principles in the literature that we organized across three domains and twelve main associated principles: i) Project management and governance domain contains five principles: a. Landscape scale, b. Prioritization, c. Legal and normative compliance, d. Participation, e. Adaptive management; ii) Human aspect domain with four principles: a. Enhance livelihoods, b. Inclusiveness and equity, c. Economic diversification, d. Capacity building; iii) Ecological Aspects domain with three principles: a. Biodiversity conservation, b. Landscape heterogeneity and connectivity, c. Provision of ecosystem goods and services. The main gaps in the literature include a lack of reference to the socioeconomic and monitoring aspects of FLR, although recently these subjects have been increasingly addressed.

  1. Materials and Methods

The answer/justification that “Our limit was 2017 because the analysis was done in early 2018” should be stated in the methods section, perhaps on line 135. It would be the second sentence of the 2nd paragraph of section 2.1.

Added the sentence where the reviewer suggested: ‘We sampled the literature until 2017 because this review was carried out in early 2018.’

Line 159-160: “...we recognize that our method may have neglected a few organizations (e.g., WWF).” --Omitting WWF as an organization is very disturbing, given the tremendous work of that organization on FLR and other restoration works across several continents and countries. Please justify that your findings are not limited, i.e., how such important omission does not affect the validity of your overall findings, and perhaps their generalization. For instance, you may state that most WWF works on FLR and restoration were captured in the peer-reviewed papers that you reviewed. This kind of appears in the first sentence of your result...So, for instance, writing the following would work. “...we recognize that our method may have neglected a few organizations (e.g., WWF), but whose works are well represented in the sample of scientific literature reviewed.”

We added the reviewer suggestion, with examples in the literature sampled of FLR practices developed by WWF: ‘…we recognize that our method may have neglected the grey literature of a few organizations (e.g., WWF), however, the work of such organizations is found in the scientific literature sampled (e.g., [27,45,46, among others]).’

  1. Results

It is good to see a reorganization of the former sections 3.3, 3.4, 3.5, 3.6, 3.7, and 3.8 under a new section ‘3.4. FLR Concepts.’ Please revise the numbering, though. Current section ‘3.4.FLR Concepts’. Should be numbered 3.3. In addition, I think it is important to revise the title as this is confusing with the previous section ‘3.2 FLR definitions, aims and associated concepts’ that discussed some concepts associated to FLR as well. A suggestion is: ‘Themes and Dimensions associated with FLR in the Literature. Another alternative (much recommended) is to avoid using the word ‘concepts’ in section 3.2 in order to keep section 3.3 as ‘FLR concepts’.

In summary, my suggestions include:

  • Changing the title of section 3.2 to ‘3.2. FLR definitions and aims in the Literature’

We changed to ‘3.2. FLR definitions and aims’, since ‘in the literature’ is implicit, as it is in the Result section of a literature review.

  • Revising the sentence in lines 229-231. You may write: “Practical translations of FLR definitions during implementation vary in certain aspects, such as how landscape is defined, how scale is incorporated (temporal and geographical), and how ecological dimensions are considered.”

Altered according to reviewer’s suggestion.

  • Renumbering section 3.4 to 3.3 and keeping the title as ‘3.3 FLR and Associated Concepts in the Literature’

We changed to ‘3.3. FLR and Associated Concepts’, since ‘in the literature’ is implicit, as it is in the Result section of a literature review.

  1. Guiding Principles

I do not see the authors’ response being reflected in the revised manuscript. On line 499, this section is still numbered as section 4, instead of section 3.5 of the Results as said.

Altered according to reviewer’s suggestion to ‘3.5. Guiding principles of Forest and Landscape Restoration’.

  1. Gaps and Recommendations

Line 662: write “...presented a thorough ....”not “...present a throughout...”

Altered according to reviewer’s suggestion.

Lines  668-671.  The sentence  needs  to  be  revised  for  clarity.  I  would  break  it  in  two parts. “Monitoring  is  highlighted  for  the  activities  encompassed  in  FLR,  such  as  forest restoration  and  income,  and  more  recent  publications  highlight  this  need  and  provide guidelines [78, 109]. Nonetheless, part of the literature sampled and reviewed here does not reinforce the need to use landscape-level monitoring indicators.”

Altered according to reviewer’s suggestion.
